# The Emerging Role of Extracellular Vesicles in the Glioma Microenvironment: Biogenesis and Clinical Relevance

**DOI:** 10.3390/cancers12071964

**Published:** 2020-07-19

**Authors:** Anjali Balakrishnan, Sabrina Roy, Taylor Fleming, Hon S. Leong, Carol Schuurmans

**Affiliations:** 1Sunnybrook Research Institute, Toronto, ON M4N 3M5, Canada; anjali.balakrishnan@mail.utoronto.ca (A.B.); or taylorfleming@rogers.com (T.F.); hon.leong@sri.utoronto.ca (H.S.L.); 2Department of Biochemistry, University of Toronto, Toronto, ON M5S 1A8, Canada; 3Independent Researcher, Montreal, QC H3G 2J7, Canada; sabrinaroy30@outlook.com; 4Department of Medical Biophysics, University of Toronto, Toronto, ON M5G 1L7, Canada

**Keywords:** glioma, extracellular vesicles, tumor microenvironment, biomarkers, liquid biopsy

## Abstract

Gliomas are a diverse group of brain tumors comprised of malignant cells (‘tumor’ cells) and non-malignant ‘normal’ cells, including neural (neurons, glia), inflammatory (microglia, macrophage) and vascular cells. Tumor heterogeneity arises in part because, within the glioma mass, both ‘tumor’ and ‘normal’ cells secrete factors that form a unique microenvironment to influence tumor progression. Extracellular vesicles (EVs) are critical mediators of intercellular communication between immediate cellular neighbors and distantly located cells in healthy tissues/organs and in tumors, including gliomas. EVs mediate cell–cell signaling as carriers of nucleic acid, lipid and protein cargo, and their content is unique to cell types and physiological states. EVs secreted by non-malignant neural cells have important physiological roles in the healthy brain, which can be altered or co-opted to promote tumor progression and metastasis, acting in combination with glioma-secreted EVs. The cell-type specificity of EV content means that ‘vesiculome’ data can potentially be used to trace the cell of origin. EVs may also serve as biomarkers to be exploited for disease diagnosis and to assess therapeutic progress. In this review, we discuss how EVs mediate intercellular communication in glioma, and their potential role as biomarkers and readouts of a therapeutic response.

## 1. Introduction: Extracellular Vesicles and Glioma 

The first hint that extracellular vesicles (EVs) might exist and have biological activity was the 1967 observation that “platelet dust” participates in clot formation [1]. This was followed in the 1980s by the use of electron microscopy to visualize EVs emanating from late endosomes in reticulocytes [2,3]. These vesicles were found to carry transferrin receptors, and were eventually termed ‘exosomes’ [4]. Seminal work in the 1990s then uncovered a role for EVs derived from B cell lymphocytes [5] and dendritic cells [6] in antigen presentation to T cells. These reports gave birth to an entire new field of research that was predicated on the notion that EVs have biological actions and are not solely ‘waste carriers’, as initially thought [5,6]. Since then, EVs have been shown to mediate intercellular communication between neighboring and distantly located cells in a vast array of biological contexts (reviewed extensively in References [7,8,9,10,11,12]). The specific roles that EVs play in intercellular communication is due to their selective and cell-type-specific loading of lipids and nucleic acids, including DNA, messenger RNA (mRNA), microRNA (miRNA) and non-coding RNA (ncRNA) [13,14]. EVs also carry protein cargo, which may be packaged into EVs non-specifically, but nevertheless serve as a readout of cellular state and can also influence the physiology of recipient cells [13,14]. Notably, encapsulating vulnerable cargo in vesicular structures renders them inaccessible from degradation by ribonucleases, deoxy-ribonucleases and proteases in the extracellular space, as these enzymes cannot traverse the EV lipid bilayer; as such, EV biogenesis has been an important evolutionary advancement that has allowed for complex intracellular signaling to take place in multicellular organisms [15,16,17]. 

All cells have unique physiologies and molecular identities, and so too do their derivative EVs. EV cargo differs depending on cell type and a cells’ physiological status and may thus be used to trace the cell of origin [14]. Moreover, as “highly-stable reservoirs of disease biomarkers” (Exocarta; http://www.exocarta.org/), EVs may serve as valuable indicators of health and disease [12]. When EVs are secreted and enter the peripheral circulation, they may contain biomarkers that carry mutational, cell signaling and microenvironmental information that could be used as readouts of a therapeutic response. Given that a current limitation in neuro-oncology is that tumor progression can only be monitored radiologically, new non-invasive measurements of disease progression could revolutionize patient care. In the context of glioma, the ability to detect the molecular state of a brain tumor in patient biofluids would significantly facilitate patient diagnosis, disease stratification and treatment monitoring in a non-invasive fashion. Early support for this notion came from the demonstration that cells derived from human glioblastoma tissue secrete EVs in vitro [17], and the content of EVs in serum and cerebrospinal fluid (CSF) (e.g., presence of amplified epidermal growth factor receptor (EGFR) in EVs in CSF of glioblastoma patients [18]) differs between patients with and without glioma [17,18,19,20].

In this review, we summarize recent literature describing the roles of EVs in mediating neural cell communication, especially in a glioma context, and describe the potential clinical utility of EVs for glioma subtyping and as biomarkers for glioma detection and therapeutic monitoring. The use of ‘humanized’ rodent models has galvanized this research, with patient-derived tumor xenografting now a mainstay approach to model glioma, including for the study of EVs in disease progression and as biomarkers.

## 2. A Brief Primer on EV Classification, Isolation and Biogenesis

### 2.1. EV Classification and Isolation

In healthy and diseased states, cells release EVs of different sizes and intracellular origins. The result is a heterogeneous mix of membranous vesicles, collectively termed “EVs”. While the definition of EV has evolved over time, the International Society for Extracellular Vesicles (ISEV) currently defines EVs as naturally released, non-replicative particles that are delimited by a lipid bilayer [21]. However, nomenclature in the field has been historically ‘muddled’, with researchers initially classifying EVs based on physical characteristics, composition and even cell of origin [21]. Terminologies such as exosomes, microparticles and microvesicles have also been used interchangeably, further complicating the field. The ISEV thus set out to standardize the naming system, now classifying EVs (the generic term) based on physical characteristics, with particles < 200 nm in diameter called small EVs (sEVs) while larger particles (>200 nm) are termed medium (mEVs; between 200 and 400 nm) and large EVs (lEVs; >400 nm) [21] (Figure 1). Several markers have been identified that label these vesicular structures, including several tetraspanins, such as CD81, enriched in sEVs, CD9 in sEVs and mEVs, and CD63 in EVs of all sizes (reviewed in Reference [9]).

EVs can be isolated from different sources (e.g., conditioned media from cultured cells, or biofluids, such as urine, serum, blood, etc.) using a growing list of protocols. EV isolation strategies include sequential ultracentrifugation, high-resolution density gradient ultracentrifugation [22], direct immunoaffinity capture using canonical EV specific antibodies (CD63,CD81,CD9) [13], immunoprecipitation using magnetic beads [23], ultrafiltration [24] and the use of numerous commercial EV isolation kits based on immunophenotypes [22,25]. As EV isolation methods have been the topic of other excellent recent reviews [26,27], they will not be further elaborated on herein. However, one important consideration is that several of the less specific EV isolation methods (e.g., sequential ultracentrifugation) also collect non-vesicular particles (exomeres, ~35–50 nm) that are not membrane enclosed, but which may contribute to the molecular and biological phenotypes associated with these preparations [28,29]. Heat shock proteins (HSP) like HSP90, HSP13, Histone H2A, H3, H4, as well as Argonaute proteins (Ago1–3) are enriched in non-vesicular particles compared to exosomes [13], which may help to identify the presence of exomeres in EV preparations.

### 2.2. Small and Medium/Large EVs

Small EVs (sEVs; diameter: <200 nm), previously known as exosomes, are derived from early endosomes that mature into multivesicular bodies (MVBs) (reviewed in References [30,31]). Within MVBs, endosomal membranes invaginate to form intraluminal vesicles (ILVs) that fuse with the cellular plasma membrane to secrete the enclosed vesicles [30,31] (Figure 1). SEVs have an approximate diameter of 40–200 nm, with densities ranging from 1.08 to 1.13 g/mL depending on the cell-of-origin [13,32]. The functions attributed to sEVs in cancer are diverse, ranging from aiding tumor progression and metastases, to promoting tumor dormancy [30,33]. The potential use of sEVs for cancer immunotherapy, nano-vaccines and as diagnostic tools is also currently being explored [34,35]. 

Medium or large EVs (mEVs or lEVs) (diameter: 200–1000 nm) were previously called microparticles, microvesicles, ectosomes and oncosomes [36]. They are generated by outward budding and fission of the plasma membrane [37]. Similar to sEVs, m/lEVs can also transfer cytosolic components to neighboring cells [17,38,39], with their contents protected from extracellular degradation by an enveloping lipid bilayer (Figure 1) [40]. m/lEVs form at specific sites in the membrane where lipids and proteins aggregate in microdomains [41]. Cytoplasmic Ca^2+^ levels are also elevated in ‘budding’ cells, promoting clustering and activation of phospholipid scramblases, which move lipids across the plasma membrane, and floppases, which transport lipids such as phosphatidylserine from the inner to the outer membrane leaflet (reviewed in References [41,42]). Notably, presentation of lipids such as phosphatidylserine on the outer membrane leaflet serves as an ‘eat me’ signal that alerts microglia in the brain (or macrophage elsewhere in the body) that a cell is under stress and should be eliminated [14,41]. Annexin A1 has recently been reported to be a distinguishing marker expressed in m/lEVs, and not in sEVs [13]. 

Large EVs also include larger apoptotic bodies (diameter: 1–5 µm) that are released from cells undergoing fragmentation due to programmed cell death (reviewed in References [43,44]). With the onset of apoptosis, cells undergo a cascade of structural changes (apoptotic cell disassembly) involving blebbing and protrusion of the cell membrane, followed by release of apoptotic bodies (Figure 1). These EVs tend to fall in the larger size range, however smaller apoptotic bodies (called apoptotic vesicles) in the size range of sEVs have also been identified [45,46,47]. Similar to other m/lEVs, apoptotic bodies present phosphatidylserine on their surfaces, so that they are quickly cleared [43,44]. 

The list of vesicle types continues to grow as methods of analysis become more sophisticated and classification criteria more granular. Other categories of EVs include autophagic EVs, released via autophagy-dependent pathways [48], and arrestin-domain-containing protein 1 (ARRDC1)-mediated m/lEVs [49,50]. In this review, we will use the updated sEV and m/lEV terminology that we apply to studies that defined vesicles as exosomes or microparticles/microvesicles, respectively. 

### 2.3. ESCRT-Dependent and ESCRT-Independent Pathways for EV Biogenesis

At least 10–15 proteins are known to play a functional role in EV formation (reviewed in Reference [9]), with the major players summarized herein. The most common sEV biogenesis route involves an endosomal sorting complex related to transport (ESCRT)-dependent pathway (extensively described in other reviews; [51,52]). Briefly, ESCRT-dependent sEV biogenesis and release involves the concerted actions of ESCRT complex proteins—ESCRT-0, I, II, III [31,53]. ESCRT-0 proteins have ubiquitin binding sites that recognize and sequester ubiquitinylated cargo in the late endosomal membrane. ESCRT-0 proteins then sequentially recruit ESCRT-I, II and III complex proteins using accessory proteins like ALG-2-interacting protein X (ALIX) (summarized in References [51,53]). Deformation of the endosomal membrane to form vesicular buds containing cytosolic cargo involves ESCRT-I and II complexes [53], while vesicle scission and release of ILVs into MVBs involves ESCRT-III complex proteins (reviewed in References [31,51,54]).

ESCRT-independent mechanisms also sort cargo into ILVs of MVBs [55,56,57,58,59]. One such mechanism involves neutral sphingomyelinase2 (nSMase2), an enzyme that acts upon sphingolipids in lipid-raft microdomains within the endosomal membrane [55]. nSMase2 hydrolyzes sphingomyelin into phosphocholine and ceramide [60]. Ceramide is enriched in sEVs and is involved in targeting cargo into ILVs and packaging of sEVs [55]. Sphingosine-1-phosphate, a ceramide metabolism by-product, also exhibits a role in sorting cytosolic cargo into ILVs [61]. Budding of ILVs is promoted by ceramide, courtesy of their cone-shaped structure that promotes membrane bending [62], which in turn induces smaller microdomains within endosomal membranes to merge into larger domains [55,63]. A reduction in nSMase2 activity using a pharmacological inhibitor, GW4869, blocks the packaging of protein [55] and miRNAs [64] into sEVs. After sEV synthesis, Rab GTPases (e.g., Rab27a, Rab27b, Rab11, Rab35) are involved in sEV secretion and recycling of proteins between the endosomal compartment and plasma membrane (summarized in Reference [65]). 

Biogenic pathways for other vesicle types are also beginning to be elucidated, such as acidic sphingomyelinase (aSMase), which is activated and recruited to the plasma membrane to promote m/lEV shedding [62,66]. Sphingomyelin enriched in the outer leaflet of the membrane is hydrolyzed by aSMase, destabilizing the membrane and facilitating m/lEV shedding [66,67]. Other regulators involved in m/lEV formation and release include Arf6, involved in endosomal recycling [68], and the small GTPase, RhoA [69]. 

## 3. Role of EVs in the ‘Healthy’ Nervous System 

Gliomas are composed of a mixture of malignant cells (’tumor cells’) as well as as non-malignant neurons and glial cells (’neural cells’), and a variety of other inflammatory and vascular cells (’stromal cells’). Tumor cells carry driver mutations causally implicated in oncogenesis, but other cells in the microenvironment also participate in tumor cell proliferation and growth. A solid understanding of EV production by ‘normal’ brain cells is required as these vesicles can impact tumor growth and progression. As published literature on brain EVs is vast, our survey is not comprehensive, but rather gives a flavor for the types of roles that EVs play in a ‘healthy’ brain (Figure 2). 

### 3.1. Neural Stem and Progenitor Cell-Derived EVs

In the adult brain, neural stem cells (NSCs) are restricted to a few neurogenic zones that repopulate specific brain regions throughout life, including the ventricular-subventricular zone (V-SVZ) in the forebrain that repopulates the murine olfactory bulb and human striatum, the subgranular zone (SGZ) that repopulates the mouse/human dentate gyrus and the mediobasal hypothalamus that gives rise to new hypothalamic neurons [70,71,72]. Outside of these niches, the adult NSC response is limited, and while some neuroblasts are produced in response to injury in other brain regions [73,74], few new neurons survive and integrate to register any meaningful recovery [75]. Nevertheless, NSC-derived EVs have been studied extensively with regards to their neuroprotective properties in injury models, such as stroke [76], and for their ability to modulate microglia activity [77]. Reprogrammed NSCs (derived from mouse fibroblasts and astrocytes in vitro) were also found to secrete EVs that promote their own proliferation by activation of Mitogen-activated protein kinase - extracellular signal-regulated kinase (MEK) / extracellular signal-regulated kinase (ERK) signaling [78]. sEVs secreted from human-induced pluripotent stem cells (hiPSCs)-derived neurons also regulate neural circuit assembly [79]. Finally, a recent study found that sEVs secreted by hypothalamic NSCs into the CSF slow down the aging process in rodent models in vivo [80]. Taken together, these studies and others support the idea that NSC-derived EVs may have therapeutic value for the treatment of brain injury or neurodegenerative disease, as reviewed elsewhere [81], and possibly for the study of longevity. With respect to glioma, brain tumor-propagating cells (BTPCs) are thought to arise from adult NSCs, and the adult NSC niche may support the growth and division of BTPCs [82]. Thus, understanding how NSCs signal to tumor cells is essential to devise strategies to block tumor cell proliferation.

### 3.2. Neuron-Derived EVs

Neuronal sEV release was initially detected in cultured rat cortical cells that received a membrane depolarization stimulus, with glutamate receptor subunits part of the sEV cargo [83]. Other studies have since confirmed that EVs are released from rat cortical neurons in vitro in response to glutamatergic activity [84]. sEV release during neuronal firing has been proposed to have a ‘waste disposal’ role, removing miRNAs to reduce their silencing effect in ‘active’ human neuronal cultures in vitro [85]. In addition, sEVs released from rat cortical neurons in cell culture are primarily taken up by neuronal and not glial cells and aid inter-neuronal communication [86]. Cultured mouse embryonic neurons form close-knit networks that uptake neuronal EVs and simultaneously re-secrete EVs (via endogenous secretory endosomes) to aid EV spreading [87], and in this manner, promote neural circuit development [79]. Recently, neuronal sEV release has been confirmed in vivo using a transgenic CD63-green fluorescent protein (GFP) mouse reporter line, which revealed that activity-dependent sEV release occurs in post-synaptic soma and dendrites [88]. Contrasting to in vitro studies, this transgenic animal revealed that neuronal EVs, carrying miR-124 in their cargo, are taken up by astroglia, resulting in reduced Glt1 expression levels [88]. Neuronal EVs have also been described as carriers of miRNAs that act non-cell autonomously in other studies [85,89,90]. For example, sEV release of miR-132 promotes vascular integrity in the zebrafish brain by targeting *eef2k* in endothelial cells [90]. Similarly, sEV release of signaling proteins in neuronal EVs have been conjectured to play a role in intercellular communication, as well as for disposal of these proteins [83,91,92,93]. Neuronal secretion of growth factors such as VEGF and FGF2 in sEVs [94] could also impact glioma growth.

### 3.3. Astrocyte-Derived EVs

Astrocytes are multi-functional macroglial cells that have a wide host of different functions, they provide structural and trophic support for neurons, contribute to the blood–brain barrier (BBB) and maintain myelin integrity [95]. Cultured cortical astrocytes were initially shown to shed m/lEVs via aSMase in response to P2X_7_ receptor-Adenosine triphosphate (ATP) ligand binding [66]. EVs derived from two-day-old rat cortical astrocytic cultures carry various growth factors (e.g., VEGF, FGF2) [96] and promote neurite outgrowth and neuronal survival in vitro [97]. The neuroprotective effects of astrocytic EVs have been studied extensively, identifying critical cargo that is transferred to neuronal cells, including neuroglobin [98], synapsin, which is released during oxidative stress and ischaemic conditions [97], and ApolipoproteinD, also released in response to oxidative stress [99]. Astrocytic EVs also promote the differentiation of rat oligodendrocyte precursor cells (OPCs) in vitro, and interestingly, as astrocytes age, this capacity declines [100]. A key study described the role of astrocytic EVs in promoting proliferation and survival of breast cancer and melanoma cells disseminating to the brain using an orthotopic mouse xenograft model system [101]. Tumor cells metastasizing to the brain were found to promote astrogliosis, resulting in the release of miR-19a-loaded sEVs from reactive astrocytes. Increased miR-19a levels in the brain microenvironment reduce *Phosphatase and tensin homolog* (*PTEN)* expression (miR-19a target), an important tumor suppressor, culminating in tumor cell growth and inhibition of tumor cell apoptosis [101]. This study demonstrated the important role that astrocytic EVs play in tumorigenesis and highlighted the importance of considering EV secretion by ‘normal’ cells in the brain tumor microenvironment when studying disease mechanism.

### 3.4. Oligodendrocyte-Derived EVs

Oligodendrocytes are myelinating glial cells in the central nervous system [102] and they release sEVs in a Ca^2+^-dependent manner, upon receiving neuronal stimuli [103,104,105,106]. Exocytosis of EVs by an oligodendrocyte precursor cell (OPC) line, Oli-neu, is brought about by Rab35 GTPase and the GTPase activating protein TBC1D10A–C [103]. sEVs released from oligodendrocytes in culture are characteristically enriched in myelin proteins (e.g., phospholipid protein (PLP), 2’, 3’-cyclic nucleotide 3’-phosphodiesterase (CNP)) [55,103,104,106,107]. Thus, EV secretion by oligodendroglial cells permits expulsion of excess myelin proteins, and importantly, plays a role in promoting neuron-oligodendrocyte communication [106]. Indeed, neuronal uptake and functional retrieval of oligodendroglial sEVs occurs at axonal and somadendritic sites in vitro [106], and influences neuronal gene expression (*Plp, Ier3, Vgf, Bdnf*), provides metabolic support and enhances neuronal activity [108]. Oligodendrocyte-derived EVs also exhibit a protective function in neurons against oxidation-induced stress or nutrient deprivation in vitro, potentially mediated by EV delivery of catalase and superoxide dismutase (SOD) 1 enzymes [106,108]. Surprisingly, oligodendrocyte sEVs inhibit oligodendrocyte differentiation in culture [107]. EVs released by Oli-neu cells are also selectively taken up by microglia via micropinocytosis, resulting in EV degradation without microglial activation [109]. OPCs cultured in close contact with astrocytes result in increased OPC-sEV release coupled with an increase in OPC proliferation regulated via integrin β4-mediated cell adhesion [110]. Thus, oligodendrocyte-derived EVs perform a gamut of functions in the neural niche and may have important actions on glioma cells as well. 

### 3.5. Microglia-Derived EVs

Microglia are specialized macrophages that mount immune responses in the brain [111]. Release of sEVs and m/lEVs from microglial cells has been extensively studied [66,112,113,114,115]. Several inducers of microglial EV release have been found, including Wnt3a [113], serotonin [115] and lipopolysaccharide (LPS), which also induce an increase in inflammatory cytokines (tumor necrosis factor (TNF) and interleukin-6) in the EV cargo [116]. Microglial EVs have actions on neuronal cells, with some examples including m/lEVs inducing neuronal activity [114] and sEVs clearing degenerating neurites to facilitate synaptic pruning in vitro [117]. Given the importance of microglia and infiltrating peripheral macrophages in facilitating glioma cell proliferation and migration [118], it seems likely that the secretion of EVs by these immune cells would have biological consequences. Indeed, EVs isolated from gamma interferon/lipopolysaccharide (IFN-γ/LPS)-stimulated microglia can reduce tumor size in a glioma mouse model [119]. The potential of microglia-derived EVs as a nano-therapy for glioma is also now being further investigated [120].

## 4. Role of Tumor-Derived EVs in Glioma

### 4.1. Glioma Subtypes

Gliomas are primary brain tumors comprised of tumor cells with gene expression profiles and morphological characteristics similar to glial cells (e.g., astrocytes, oligodendrocytes), as recently reviewed [121,122,123]. Glioma subtypes include oligodendroglioma, astrocytoma, glioblastoma, ependymoma, schwannoma and neurofibroma [121,124]. Depending on the speed and extent of tumor growth, histological features and tumor invasiveness, gliomas have been graded by the World Health Organization (WHO) between Grades I and IV [125,126]. Low-grade gliomas (Grade II–III) include astrocytomas and oligodendrogliomas [126,127], while the most common type of higher grade glioma is an astrocytoma known as glioblastoma (GBM) [124]. 

As the most aggressive and common glioma, GBM deserves special mention. To identify clinically relevant subtypes, The Cancer Genome Atlas (TCGA) performed a multi-platform analysis to generate a robust gene expression-based molecular classification of GBM [128,129]. This landmark classification tool identified four distinct molecular subtypes for GBM (classical, mesenchymal, pro-neural and neural) and demonstrated that subtypes correlate with clinical phenotypes and treatment responses [128]. More recently, classification of three distinct forms of GBM of the classical, pro-neural and mesenchymal subtypes has been proposed, discarding the neural subtype as a signature that was likely associated with contaminating mRNA from non-tumor cells [50,130]. Classical GBMs frequently have copy number alterations and/or mutations in EGFR that lead to its overexpression and activation [129]. Pro-neural GBMs harbor high-level amplification and/or rearrangements of Platelet-derived growth factor receptor alpha (PDGFRA) that render it constitutively active [129]. However, not all pro-neural GBMs harbor PDGFRA alterations—some feature isocitrate dehydrogenase 1 or 2 (IDH1/IDH2) mutations [129]. However, IDH mutations are only detected in secondary GBMs (arising from grade II and III gliomas), which constitute ~5% of the GBM cohort [131,132,133]. Finally, mesenchymal GBMs are characterized by NF1 loss (summarized in Reference [134]). 

While the above profiles suggest uniformity amongst GBM subtypes and the cells contained therein, tumor composition is complex. There is additional heterogeneity as proliferating tumor stem and progenitor cells undergo lineage progression and differentiation, and the molecular identity of tumor cells can evolve over time, with pro-neural signatures often resolving into more mesenchymal, aggressive tumor phenotypes [135]. Accordingly, single cell analyses revealed that cells with different subtype-specific gene expression signatures are found within individual tumors in different proportions [136]. Understanding how glioma cells interact with cells in the microenvironment is an essential step in understanding disease progression. Of note, analysis of the vesiculome for each GBM subtype has revealed differences in key EV pathway components [135]. This section delves into the key studies focused on the role of EVs derived from GBMs and oligodendrogliomas.

### 4.2. EVs in Glioblastomas

GBMs are Grade IV astrocytomas and represent the most lethal brain tumor subtype [124]. NSCs residing in the SVZ carrying driver mutations (*TERT* promoter mutation, *EGFR, PTEN* and *TP53* mutation) have been reported as the cell of origin for GBM [137]. However, other studies have found that any type of neural cell, including hippocampal NSCs [138], adult neurons and astrocytes [138,139], OPCs [140,141] and OPC intermediates (expressing low *Pdgfra* and high *Olig1/2* levels) [142], can also give rise to GBM-like tumors. Interestingly, human GBM cell-derived EVs can drive transformation of human NSCs towards a tumorigenic state in vitro, highlighting the potential importance of GBM EVs in tumorigenesis [143]. Supporting the importance of cell–cell interactions in glioma, the requirement for two oncogenes to drive tumor formation (i.e., *RasV12, scribbled*) can be achieved even when these genes are not expressed in the same cells, but rather in neighboring clones [144]. Since then, several studies have revealed the role that EVs play in mediating cell–cell interactions in glioma, as elegantly reviewed [145,146]. 

To give an overall appreciation for the importance of EV communication in glioma, we highlight a handful of key studies. The bioactive nature of GBM EVs is mediated by the inclusion of tumorigenic proteins (EGFRvIII), DNA (mitochondrial DNA) and RNA (e.g., *Annexin A2* mRNA, miRs, including miR-10b, miR-21, miR-221) in EV cargo [147,148,149,150,151,152,153,154]. For instance, a long non-coding RNA (lncRNA) antisense transcript of hypoxia-inducible factor-1α (AHIF) is found in human GBM cell line-derived sEV cargo, which increases viability and invasive properties of GBM cells in vitro [155]. Notably, while RNA transcripts are carried uniformly by both sEVs and lEVs, protein cargo amounts are significantly higher in GBM cell-derived lEVs compared to sEVs, including EGFRvIII [146,149]. Apart from EGFRvIII, EVs released by GBM cells (including GBM cell lines and GBM patient-derived glioma/cancer stem cell line (GSC/CSC)) were demonstrated to carry the chloride intracellular channel-1 (CLIC1) protein [156]. CLIC1 plays a role in cell cycle regulation [157] and has previously been implicated in GBM growth, such that high CLIC1 levels correlate to poor prognosis in GBM patients [158]. It was later demonstrated that treatment of GBM cells with EVs carrying CLIC1 (1 µg/mL) resulted in increased GBM cell proliferation in vitro and in a mouse GBM xenograft model system in vivo [156]. 

High expression of miR-21 is a common feature in GBM patient tissues and established GBM cell lines in vitro, and suppression of mi-R21 in vitro results in decreased proliferation and increased apoptosis [159,160]. Additionally, GBM patients with high miR-21 levels exhibited poor prognosis [160]. Interestingly, miR-21 carried in sEVs in the CSF of GBM patients has also been reported, where the level of miR-21 in sEVs is directly related to the glioma status of the patient [154]. Thus, recurring GBM patients demonstrate low levels of sEV miR-21 post-surgical resection, compared to higher levels of miR-21 observed prior to surgical resection [154]. Hence, sEVs carrying miR-21 in the CSF are an excellent biomarker for checking the glioma status of GBM patients. Also, antisense miRNA oligonucleotides against miR-21 loaded onto sEVs have been recently tested as a delivery system in vivo using a mouse xenograft model for assessing their therapeutic potential [161]. On similar lines, miR-21 inhibition in GBM cells in vitro was introduced using sEVs engineered to carry an miR-21 sponge construct (i.e., three miR-21 complementary sequences joined by linker sequences) [162]. Using this approach, suppression of miR-21 target genes, *PDCD4* and *RECK,* in GBM cells was reverted along with an increase in apoptosis and decrease in cell proliferation, similar to prior studies [159,162]. Additionally, the introduction of sEV-loaded miR-21 sponge constructs in a rat xenograft model of GBM led to a significant reduction in tumor volume compared to the control group [162]. Thus, miR-21 inhibition via sEVs is an active area of research for developing new therapeutic strategies. 

Tumor repressive functions have also recently been attributed to GBM-derived EVs. The miR-302-367 cluster was found to repress stemness in GBM patient-derived GSC lines [163]. Fareh et al. engineered GBM patient-derived GSC lines to express the miR-302-367 cluster, which then released sEVs carrying miR-302-367 as cargo. Transfer of miR-302-367 to recipient GSC lines via sEVs repressed the stem cell-like nature of GSCs, as demonstrated via an inhibition of cell stemness and proliferation marker expression (e.g., Shh, SOX2, Cyclin D, Cyclin A) [163]. Thus, miR-302-367 can block GBM growth in a paracrine fashion and miR-302-367 delivery via sEVs is currently being assessed as a potent therapeutic approach for GBM patients. Interestingly, human GBM cell-derived sEVs were reported to carry *O-methylguanine-DNA methyltransferase* (or *MGMT*) mRNA, which is an indicator of GBM drug resistance status [164]. Additionally, temozolomide (TMZ)-resistant human GBM cells release sEVs carrying the lncRNA SBF2-AS1, which represses miR-151a-3p and XRC44 expression in vitro [29]. Transfer of lncRNA SBF2-AS1 via sEVs to neighboring GBM cells also endowed TMZ resistance in the recipient GBM cells [29]. Thus, lncRNA SBF2-AS1 in sEVs can be an excellent readout/biomarker of TMZ resistance in GBM patients. Thus, personalization of chemotherapeutic treatment may be possible using EV cargo as more precise readouts of the current glioma status [164].

Numerous studies have reported the non-cell autonomous effects of GBM tumor-derived EVs on neural cells present in the tumor microenvironment, including astrocytes, endothelial cells and pericytes (Figure 2) [145,146]. Some key studies are worth mentioning due to their emerging role as biomarkers for clinical staging and treatment response. Al-Nedawi et al. found that oncogenic EGFRvIII secreted by GBM EVs is taken up by endothelial cells, which are reprogrammed to express VEGF, and activate VEGF receptor 2 (VEGFR2) [165]. GBM sEVs are also enriched in long non-coding RNA activated by transforming growth factor (TGF)-β (lncRNA-ATB), which is transferred to astrocytes, where it suppresses miR204-3p to increase Glial fibrillary acidic protein (GFAP) expression, leading to reactive astrocyte activation and enhanced glioma invasiveness [166]. Additionally, GBM EVs modulate gene expression in astrocytes, reducing expression of the tumor suppressor gene *TP53*, and elevating the expression of oncogenic proteins such as Myc proto-oncogene protein (MYC) [167]. Notably, while most of these studies have used in vitro modeling, in vivo transfer of material between glioma cells and ‘normal’ brain tissue has recently been demonstrated using a Cre/LoxP reporter system [168]. EV-mediated GBM interactions with cells in the microenvironment are reciprocal in nature. For example, endothelial cell-derived EVs isolated from a GBM mass promote glioma cell migration [169]. Similarly, fibroblasts associated with GBM secrete EVs that are taken up by tumor cells to promote glycolysis [170]. 

Finally, attempts have been made to test the functional role of EV secretion in gliomagenesis by knocking-out/-down various drivers of EV biogenesis and secretion that are expressed in gliomas [135]. Recent knockdown studies have reported that Rab27a/b regulates tumor growth and EV secretion in a mouse glioma model [101,168]. Thus, further studies focusing on other drivers of EV biogenesis will aid in elucidating the role of EVs in gliomagenesis. 

### 4.3. EVs in Oligodendroglioma 

Oligodendroglioma (ODG) are slower growing, lower grade gliomas associated with a distinct constellation of mutations, including gain-of-function mutations in IDH1/2, and chromosomal co-deletion of 1p/19q [124], with mutation of Capicua transcriptional repressor (CIC) in the retained chromosome also prevalent [171,172,173,174]. Patient-derived ODG cell lines secrete EVs [175], but only a handful of studies have examined their roles in ODG tumorigenicity [176,177]. Strikingly, instead of the growth-promoting effects largely attributed to GBM EVs [145,146], EVs derived from a mouse G26/24 ODG cell line have cytotoxic effects on neurons and to a lesser extent, astrocytes in vitro [176,177]. Two pro-apoptotic proteins were identified in G26/24 ODG-EVs that contribute to their cytotoxic effects—Fas ligand (FasL) and tumor necrosis factor-related apoptosis-inducing ligand (TRAIL) [176,177]. Interestingly, FasL was absent in the EVs derived from an anaplastic ODG patient-derived cell line UPN9333 [175], highlighting the heterogeneity of EV content. In another study, the cytotoxic effects of sEVs derived from Human Oligodendroglioma (HOG) cells on other ODG cells in vitro were attributed to the ceramide that makes up these vesicles [178]. ODG m/lEVs derived from G26/24 ODG cells were also reported to carry aggrecanases (Adamts1, Adamts4, Adamts5) in their cargo, bringing about degradation of aggrecan-rich extracellular matrices to promote tumor cell invasiveness in vitro [179]. There is thus growing support for ODG EVs exerting a potent cytotoxic effect on neurons, astrocytes (non-cell autonomous interaction) and neighboring ODG cells (cell autonomous interactions) via various means. 

## 5. Clinical Potential of EVs for Glioma: A Non-Invasive Biopsy of the Brain and Its Resident Tumor

Diagnosis of any abnormal growth within the cranium is dependent on the histological examination of biopsied tissue. Needle biopsies are invasive procedures that are particularly challenging to obtain from brain tumors, especially when tumors are not immediately accessible at the brain surface as normal brain tissue can be damaged in the needle path. A non-invasive means to understand the underlying biology of the mass in question would significantly benefit the patient, sparing them side-effects from the biopsy procedure. EVs released by the tumor that are present within the blood circulation incorporate various macromolecules that could serve as biomarkers of tumor status [180,181]. Indeed, several EV-based “liquid biopsies” have been patented and commercialized (previously reviewed in Reference [182]) with the goal of improving diagnosis and clinical workup of patients with suspected GBM. The rationale and effectiveness of EV-based “liquid biopsies” will be discussed below. 

### 5.1. A “Liquid Biopsy” for Glioma: Glioma-Derived EVs as a Biomarker Platform 

The term “liquid biopsy” or “fluid biopsy” is a neologism that refers to an alternative to needle biopsy for obtaining the same information regarding a disease or lesion via analysis of a biofluid [183]. In oncology, a “liquid biopsy” is typically the analysis of a biomarker (nucleic acid, protein, metabolite) released by a tumor and its abundance in human biofluids. The biomarker of interest may provide information regarding grade (indolent versus aggressive), stage (size and local spread beyond primary site), tumor biology (increased versus decreased cell proliferation, e.g., EGFRvIII is associated with increased proliferation of glioma cells), prognosis (increased versus decreased survival, e.g., IDH1 mutation is an independent factor for longer survival in GBM patients compared to patients with wild-type IDH1) and treatment response information, when analyzed before and after a chosen therapy (surgery, radiation therapy, systemic therapy) [183,184]. 

The brain is an immune-privileged organ and protected by the BBB, which is a vascular wall that is impermeable to most cells, drugs, metabolites and proteins [185]. “Liquid biopsies” based on the analysis of whole blood are based on the premise or assumption that the BBB within a brain tumor may be physically compromised and hence permissive to release or intravasation of biomarker material (proteins, cells, metabolites, DNA/RNA, extracellular vesicles, etc.) into the hematogenous circulation. Indeed, Zhao et al. [186] determined that BBB permeability exists in GBM and can vary with stage of the tumor, but not all GBM tumor vasculature is compromised. Pockets or regions of BBB permeability were found to correlate with intravasation of GBM-EVs and subsequently elevated in peripheral whole blood samples [186]. Established GBM biomarkers (e.g., mutated EGFRvIII) are commonly used due to its robust expression in the majority of GBM tumors and cell surface expression in GBM cells. Significant literature describing utility of EGFRvIII as a GBM-specific EV marker has led to its use as a clinical diagnostic tool [187] and as a prognostic tool [188]. At the time of initial patient presentation where the biology of the suspicious mass is not yet determined, GBM-EVs may also provide risk-stratifying information [189,190].

Currently, tissue-based biopsy is typically performed at the time of surgical resection. However, a resection may not be planned in patients with deep-seated lesions or with significant co-morbidities. An EV-based “liquid biopsy” would therefore allow clinicians to avoid surgery in poor surgical candidates and instead proceed directly to radiation and/or chemotherapy. Additionally, GBM-derived EVs may permit pre-surgical stratification of patients likely to benefit from aggressive resection. For some GBM cases, there may be a benefit in knowing the molecular phenotype prior to surgery. For example, recent reports suggest a differential advantage in achieving a gross total resection in IDH mutant GBM, compared to wild-type IDH tumors [191]. Pre-operative diagnosis of IDH mutant GBM via a “liquid biopsy” would permit a more aggressive pre-operative and intraoperative surgical plan to achieve gross total resection. An RNA-based “liquid biopsy” may also offer an improvement to current challenges over detection of the mutations on tumor tissue due to intra-tumoral heterogeneity [191]. 

RNA biomarkers found within GBM-EVs offer the greatest promise for clinical impact due to the potential low cost of their analysis and their stability in the serum when enclosed in EVs [187]. Targeting RNA biomarkers is also promising given the sensitivity of nucleic acid amplification technologies (digital droplet Polymerase chain reaction (PCR), quantitative PCR, etc.) and their abundance compared to cell-free DNA [192]. This is because a single cell will generally have two copies of each mutated genome (diploid) but will have thousands of copies of mutated RNA species which will become “cell-free” or resident within an EV when a cancer cell dies.

CSF may be the most relevant biofluid for a “liquid biopsy” for GBM and other brain cancers in order to deliver clinically relevant information for brain cancer patients [180,193]. CSF is the fluid that bathes and cushions the central nervous system, which includes the brain within the skull and spinal column [194]. In terms of total fluid volume, humans have 140–270 mL of CSF, which is considerably less than the total volume of whole blood in humans (4.5–5.5 L). This fluid is predominantly acellular and is proteomically distinct from whole blood, which makes analysis and processing more straightforward [194]. However, it suffers from one major drawback: it can only be obtained via lumbar puncture, a technically challenging means of biofluid collection and is seldom performed in brain cancer patients for diagnostic purposes. Various key studies [18,193] have enumerated brain cancer-derived EVs in CSF that express EGFRvIII, resulting in a CSF-based “liquid biopsy” that exhibits promising performance test characteristics (61% sensitivity, 98% specificity) [18]. Detection of wild-type and mutated *IDH1* mRNA, which is a prognostic factor for GBM, in CSF resulted in a “liquid biopsy” with 63% sensitivity for GBM diagnosis [18]. Hence, CSF analysis for EV-based biomarkers has yielded very promising “liquid biopsies” for GBM and other brain cancer patients that may be used clinically. 

### 5.2. Characterization of GBM Subtypes Based on Protein Biomarkers

As discussed previously, GBM is the most aggressive and common primary brain tumor with a high likelihood of recurrence post treatment [195]. The primary method of follow-up monitoring, magnetic resonance imaging (MRI), is not capable of providing information regarding the tumor’s molecular genetic alterations as it responds to treatment [196]. Currently, diagnosis is obtained from the tumor via surgical resection and tissue biopsies. Obtaining GBM tissue for analysis is invasive and associated with some degree of risk, a problem that compounds if biopsies are continually repeated [72]. To date, no circulating biomarkers in plasma are approved to diagnose GBM. Several studies have demonstrated that total EV RNA isolated from human biofluids (e.g., CSF) can be used to detect GBM-related genomic alterations [17,18,154,197,198], which collectively act as an important proof of principle. These efforts underscore the need for a plasma-based GBM signature or “liquid biopsy” as a clinically important non-invasive strategy for GBM diagnosis.

As research continues to shed light on the clinical course and subtypes of GBM, the potential of tumor-derived EVs in diagnosing, stratifying, or predicting disease has become very attractive to both clinicians and researchers [199]. In fact, investigators are actively trying to understand which protein markers can be included to detect all types of GBM via “liquid biopsy”. Researchers are encouraged to review the existing literature and large cancer datasets (i.e., TCGA) for potential EV-based GBM biomarkers. Future studies should also include putative cancer markers in the context of GBM.

### 5.3. Future Directions and Unmet Needs for a GBM “Liquid Biopsy”

The performance test characteristics of any “liquid biopsy” will rely heavily on a well-characterized true negative disease patient cohort. This would ideally be comprised of patients that are age-matched to the disease of interest, are MRI scan-negative and are at an appropriately matched female:male ratio. Since brain tissue biopsies are not performed in healthy patients, a pathological diagnosis of no GBM disease is not possible; therefore, a radiologic assessment is the next best-case scenario. If a CSF-based “liquid biopsy” is being developed, the chances of obtaining CSF via lumbar puncture from healthy volunteer patients is low, resulting in a “liquid biopsy” that may exhibit a poor set of performance test characteristics (low area under curve (AUC), low positive predictive value (PPV), low negative predictive value (NPV), etc). On the other hand, a whole blood/serum/plasma/urine based “liquid biopsy” would yield a highly feasible healthy volunteer patient cohort for assessing the accuracy of any putative GBM “liquid biopsy”. Hence, a well-characterized true negative disease patient cohort is key for the future development of “liquid biopsies” specific for any subtype of brain cancer. 

Standardization of techniques and workflow for a GBM “liquid biopsy” that relies on the analysis of plasma/serum/urine has been previously established by ISEV guidelines [21], but fewer guidelines exist for analysis of CSF samples [200,201]. Key considerations unique to CSF must be focused on the expected levels of various canonical EV biomarkers in healthy volunteer CSF samples. While these “housekeeping” biomarkers already exist for EV analysis in plasma/serum/urine, it is not clear what biomarkers should be used for CSF [202,203]. More research will eventually lead to the development of CSF-relevant guidelines by ISEV. 

Lastly, templates specific for clinical trials or clinical studies designed to validate any GBM- “liquid biopsy” for diagnosis, prognosis, risk stratification or treatment response(s) in patients would be of immense value for translational cancer researchers. These templates would establish patient inclusion/exclusion criteria, disease characteristics, clinical setting, comparator readouts/blood tests being used in the clinic, clinical imaging criteria, recommended clinical follow-up timeframes and frequency of testing. These templates would accelerate discovery and clinical validation of any promising GBM “liquid biopsy”, thus minimizing confusion and mismanagement of resources. 

### 5.4. Analysis of EVs on a Single Event Level

Next generation approaches to EV analysis will allow the full potential of EVs as glioma biomarkers to be realized. The field of EVs is rapidly evolving from one-dimensional (1D) analyses to analytical outputs that provide multiparametric information for any single EV. An example of multi-parametric analysis is nanoscale flow cytometry (nFC) of EVs, which is analogous to flow cytometric analysis of cells/EVs [204,205]. Previously, 1D analyses of EVs with dynamic light scattering (DLS) instruments such as Nanosight provided sizing information regarding all EVs present within a sample [204]. However, this method has slowly grown out of favor due to its limitations and artefacts [204]. Firstly, there is a need to discriminate between a true EV event versus noise or a non-EV event. This ideally would be based upon the surface expression of an EV marker, such as CD9/CD63/CD81, which was not possible with DLS/Nanosight. Secondly, EVs are seldom spherical and smooth on their surface, which are two key assumptions for sizing analysis of events in DLS/Nanosight techniques. Light will refract differently if there are topographical irregularities, such as DNA decorated over the EV surface, or if there are exposed membrane fragments on the EV. Hence, the sizing information collected via DLS/Nanosight is limited and lacks fidelity. Western immunoblotting is another 1D analytical technique that has been heavily relied upon for understanding the composition of EV cargo if the target EVs are amenable to purification. However, these analyses do not provide information on whether the target EVs simultaneously express multiple markers or cargo constituents, it merely confirms the presence of EV markers/cargo in that pool of EVs. The desire to understand the proportion of target EVs present in a biofluid amongst other EVs has led the field to single event EV analysis. 

Modalities such as nFC are able to provide high-throughput multi-parametric analysis of single EVs, provided that the markers being used for analysis are extracellular and also detectable via fluorophore pre-conjugated antibodies such as anti-human CD9 antibodies (mouse immunoglobulin G (IgG) 1) conjugated to a fluorophore such as Fluorescein isothiocyanate (FITC)/R-phycoerythrin (R-PE)/Allophycocyanin (APC) [184,205]. In some cases, fluorophore-based dyes such as SytoRNA or Hoechst can be used to quantitate nucleic acid content within any subpopulation of EVs. Such analyses suffer from not having an irrelevant binding control version of that dye, a control capability provided by non-immune pre-conjugated mouse IgG antibodies for reagents such as anti-human CD9 antibodies. The main benefit of utilizing nFC for single event EV analysis is that event analysis rates of >1,000 events/sec is achievable, thus providing sizing and biomarker expression simultaneously for each EV [206]. This allows the user to define and quantitate the relative proportion of specific subpopulations within a complex biofluid composed of various EV subtypes, such as plasma or urine. The multiparametric nature of flow cytometry datasets is also highly amenable to machine learning methods, in which the combination of parameters being analyzed in themselves allows for development of analysis algorithms. For example, a combination of specific EV subpopulations can be assigned to a certain disease condition, i.e., # (number) of EVs co-expressing biomarker X + biomarker Y at a certain size range (<200 nm in diameter) combined with # of EVs co-expressing biomarker X + biomarker Z at a certain size range (>800 nm in diameter). This agnostic combinatorial parameter approach eludes conventional approaches (i.e., # of EVs co-expressing biomarker X only) and holds significant potential in deriving diagnostic value from pre-existing datasets [206]. 

Immunofluorescence imaging of single EVs using super resolution microscopy is also emerging as a key technique and its main advantage is revealing the spatial distribution of a target molecule within or around an EV of interest. However, the main limitation is that EV markers do not have counterstain equivalents that are consistently utilized in immunofluorescence imaging of adherent cells. For example, Hoechst/ 4′,6-diamidino-2-phenylindole (DAPI) dyes are used to counterstain the nuclei of adherent cells which is important in locating cells of interest and for determining the focal plane prior to image acquisition. To label the cell boundary, proteins conjugated to fluorescent dyes such as phalloidin-rhodamine are used to label the cell cytoskeleton, which allows the user to understand the maximal cell volume for which the target molecule may be distributed or compartmentalized. In contrast, EVs lack these counterstains primarily because EVs do not universally have a nucleus or have nuclear content, and not all EVs contain an actin cytoskeleton that is commonly recognizable. Hence, understanding where a target molecule is within or on an EV is currently not achievable because the EV physical boundaries are not reliably definable during immunofluorescence imaging. Fluorescent-negative stains such as high-molecular weight dextrans conjugated to fluorophores could be used to image EVs via super resolution microscopy in a similar manner to how osmium tetroxide is used as a negative stain for imaging EVs via transmission electron microscopy.

Emerging on the forefront is the use of mass cytometry to perform a highly multi-parametric analysis of subpopulations of EVs in complex biofluids such as plasma. Mass cytometry differs from conventional flow cytometry in that the use of rare earth metal ions allows for >30 parameters to be used as opposed to the smaller set of fluorophores used in conventional flow cytometry (2–7 possible fluorophore dye combinations). The lack of background or “bleed through” with rare earth metal ions as the “fluorophore” in mass cytometry also offers a significant advantage in that if a rare earth metal signal is detected on an event, there is absolute certainty that it is a true signal and not an artefact [207,208]. Fluorophores suffer from noise and spectral overlaps between the dyes commonly used in flow cytometry, thus introducing doubt in any signal observed on an event, whether it be a cell or EV. The opportunity to use mass cytometry for EV analysis means that a very high number of EV subpopulations can be enumerated in a given sample, allowing for quantitation of highly specific subsets of EVs or quantitation of all possible blood cell EV subtypes in any plasma sample [209]. This would be useful in investigations profiling all immune cell EVs present in a patient plasma sample in the context of brain tumors, and other pathological conditions.

## 6. Conclusions

The importance of EVs in the healthy brain and glioma microenvironment is beginning to be appreciated. To understand how EVs contribute to glioma growth, one must also study how these vesicular structures mediate cell–cell communication between neural and non-neural cells in the brain, as these processes can be co-opted to promote glioma cell growth. Notably, investigations into EV function have not only improved our foundational knowledge of glioma biology, but also uncovered their potential to be used for “liquid biopsies”, as biomarkers to detect and identify glioma subtypes and disease progression. The further development of these liquid biopsy tools will dramatically improve diagnostic avenues as well as patient prognosis. With ever improving technologies allowing efficient EV isolation (e.g., high-resolution density gradient ultracentrifugation, direct immunoaffinity capture methods) and analysis of EVs at a single event level, the use of EVs in the clinic as diagnostic biomarkers and in therapy may soon be a reality.

## Figures and Tables

**Figure 1 cancers-12-01964-f001:**
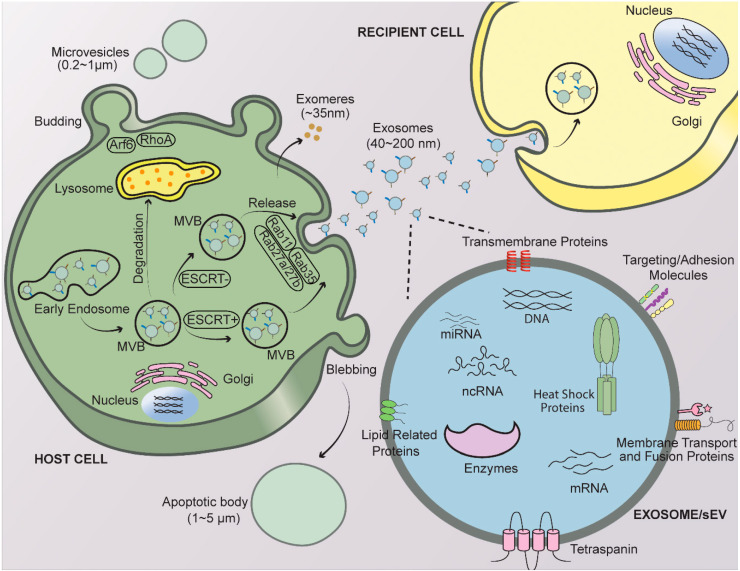
Schematic of extracellular vesicle (EV) biogenesis and signaling between neighboring cells. Generation of EVs from a host cell, depicting small (sEV; 40–200 nm), and medium to large (m/lEV; 0.2–1 µm) EVs and apoptotic bodies (1–5 µm). sEVs find their origin in the early endosomal compartment which give rise to multivesicular bodies (MVBs). MVBs fuse with the plasma membrane to secrete sEVs. Biogenesis of sEVs can occur via the endosomal sorting complex related to transport (ESCRT)-dependent pathway or ESCRT-independent pathway. Release of sEVs involves proteins such as RabGTPases (e.g., Rab11, Rab35, Rab27a/27b). Cargo in MVBs fated for degradation are processed via lysosomes. m/lEVs are released by budding of the plasma membrane regulated by proteins including Arf6 and RhoA GTPase, while dying cells undergo blebbing and form cell protrusions to release apoptotic bodies. Exomeres (~35 nm) are also released by cells via unknown mechanisms. A magnified sEV is presented on the right, demonstrating commonly found sEV cargo (DNA, messenger RNA (mRNA), microRNA (miRNA), non-coding RNA (ncRNA), long non-coding RNA (lncRNA), proteins, enzymes, heat shock proteins). Recipient cells internalize sEVs via endocytosis.

**Figure 2 cancers-12-01964-f002:**
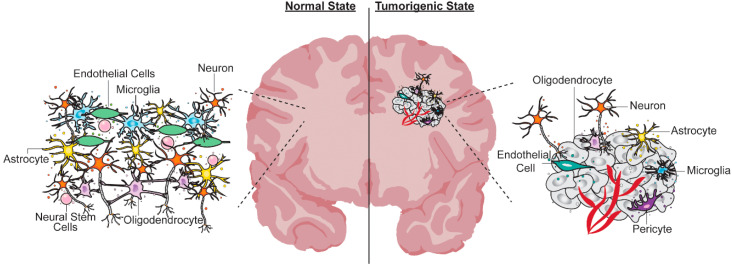
Schematic of EV-mediated cell–cell interactions in the healthy and tumorigenic brain. Normal brain (left panel) and brain tumor microenvironment (right panel). In a healthy state, EVs released by normal neural cells (neural stem cells (NSCs), neurons, oligodendrocytes, astrocytes), inflammatory cells (microglia) and endothelial cells promote cell–cell interactions, which contribute toward the formation of an intricate neural network (left side; magnified panel). In a tumorigenic state, the tumor microenvironment is a complex ecosystem of tumor cells intricately knit together with ‘normal’ brain cells. Tumor–tumor homotypic interactions and tumor–‘normal’ cell heterotypic interactions can be mediated via EVs secreted in the tumor microenvironment. Cell-specific EVs are represented as small spheres with the same color as the cell that releases them.

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
