# Peer review of "The Emerging Role of Extracellular Vesicles in the Glioma Microenvironment: Biogenesis and Clinical Relevance"

_cancers, 2020, doi:10.3390/cancers12071964_

Round 1
Reviewer 1 Report
This is an excellent review that explains clearly many aspects of EV biogenesis and analyzes the role of EVs in glioma progression and their potential use as biomarkers in liquid biopsies. The article is very well written and it is read fluently and without losing interest. I consider the authors have made a great job.
Minor corrections:
Figure 2: the font size is too small and it is difficult to read. I suggest that instead of indicating both the cell type (e.g. pericyte) and the associated EVs (e.g.pericyte-EVs) the authors could name only the cell type (with a higher font size) and then in the legend they could mention that the EVs are represented as small circles with the same colour than the cell that releases them.
Line 319: “NSCs residing in the SVZ carry driver mutations…” This is not correct. Maybe the authors meant that NSCs carrying driver mutations have been reported to be the cell of origin in GBM. Please modify.
Line 341: Al-Nedawi
Lines 342-343: please check whether this is the correct citation. I think effects on coagulation are no reported in the cited paper.
Line 350: “…are reciprocal”
Lines 393-403: please correct the line spacing
Lines 430-432; please try to rewrite these sentences, there are three “due” and sounds confusing
Line 447: a parenthesis is missing
Line 459, I do not understand the meaning of this sentence, please rewrite
Line 506, a space is missing
Line 1027: “…. Submitted work.
Author Response
Please note we have made minor revisions to the abstract for clarity (highlighted) and addressed the reviewer’s comments. All changes are highlighted in the revised review and described below:
Figures:Both Figure 1 and Figure 2 were replaced with larger font size and changes to labels.
Reviewer 1:
(1) Figure 2: the font size is too small, and it is difficult to read. I suggest that instead of indicating both the cell type (e.g. pericyte) and the associated EVs (e.g.pericyte-EVs) the authors could name only the cell type (with a higher font size) and then in the legend they could mention that the EVs are represented as small circles with the same colour than the cell that releases them.
Response: The figure has been modified as suggested, and the figure legend text has been modified to include the description of EVs (line #194-195).
(2) Line 319: “NSCs residing in the SVZ carry driver mutations…” This is not correct. Maybe the authors meant that NSCs carrying driver mutations have been reported to be the cell of origin in GBM. Please modify.
Response: The text has been modified (line #324-326) to state: “NSCs residing in the SVZ carrying driver mutations (TERT promoter mutation, EGFR, PTEN and TP53 mutation) have been reported as the cell of origin for GBM [135]. “
(3) Line 341: Al-Nedawi is incorrectly spelt.
Response: Al-Nedawi has now been correctly spelt (now line #386).
(4) Lines 342-343: please check whether this is the correct citation. I think effects on coagulation are no reported in the cited paper.
Response: The statement and reference have both been corrected. The sentence now reads: “Al-Nedawi et al. found that oncogenic EGFRvIII secreted by GBM EVs is taken up by endothelial cells, which are reprogrammed to express VEGF, and activate VEGF receptor 2 (VEGFR2) [165].” (line #386-388).
- Al-Nedawi, K.; Meehan, B.; Kerbel, R.S.; Allison, A.C.; Rak, J. Endothelial expression of autocrine VEGF upon the uptake of tumor-derived microvesicles containing oncogenic EGFR. Proc Natl Acad Sci U S A 2009, 106, 3794-3799, doi:10.1073/pnas.0804543106.
(5) Line 350: “…are reciprocal”
Response: The text has been modified to “EV-mediated GBM interactions with cells in the microenvironment are reciprocal in nature.” (line #394-395).
(6) Lines 393-403: please correct the line spacing
Response: The line spacing error has been corrected (line #437-446).
(7) Lines 430-432; please try to rewrite these sentences, there are three “due” and sounds confusing
Response: The sentences have been rewritten to: “RNA biomarkers found within GBM-EVs offer the greatest promise for clinical impact due to the potential low cost of their analysis and their stability in the serum when enclosed in EVs [187]. Targeting RNA biomarkers is also promising given the sensitivity of nucleic acid amplification technologies (digital droplet PCR, quantitative PCR, etc.) and their abundance compared to cell-free DNA [192]. (line #473-477).
(8) Line 447: a parenthesis is missing
Response: The text has been modified to include the parenthesis (line #490).
(9) Line 459, I do not understand the meaning of this sentence, please rewrite
Response: The sentence in question was “Obtaining GBM tissue for analysis is invasive and associated with some degree of risk, which compounds if repeated [175]." We have modified this to “Currently, diagnosis is obtained from the tumor via surgical resection and tissue biopsies. Obtaining GBM tissue for analysis is invasive and associated with some degree of risk, a problem that compounds if biopsies are continually repeated [200]. “(line #499-501).
(10) Line 506, a space is missing
Response: The text has been modified to correct the spacing issue (line #549).
11) Line 1027: “…. Submitted work.
Response: We have removed our unpublished work as this review will be published before the manuscript is accepted.

Reviewer 2 Report
The authors submit a well-written, clear and interesting review of the current knowledge on the role of extra-cellular vesicles in glioma growth. It includes well-defined parts starting with a rappel of the historical landmarks in the recognition of the existence of EV, description of the present technics used to isolate and characterize them as well as future technical improvements, a synthesis of the knowledge on EV biogenesis and their role in the normal brain and in glioma. The review ends on an interesting discussion on EV potential use for clinical diagnosis and follow-up of patients, with meaningful considerations to achieve such a goal.
Specific comments
- A schematic representation of the molecular pathways involved in EV biogenesis (described in paragraph 2.3) would be welcome to complete figure 1.
- The reference list contains numerous reviews. Although the authors justify their choice to refer to previous reviews in several instances throughout the article, they fail to do so in several other instances. In those cases, the references of the articles reporting the original findings have to be added, whenever possible.
- References 121 and 122 cited in regard of glioma gene expression profiles (lines 286-287) are 20 year-old. Tremendous progress having been achieved during the past years, more recent reviews have to be cited.
- Line 126, the authors wrote: “some (pro-neural GBMs) feature IDH1/2 mutations and a hypermethylator phenotype”. In GBMs, IDH1/2 mutations are detected only in so-called secondary GBMs that evolve from lower grade lesions. Secondary GBMs represent a minority of GBMs. This point should be clarified.
- The authors refer to tumor cells in glioma as “malignant glial cells” (lines 13, 147). The lineage identity of the cell(s) at the origin of glioma remains unknown, and is still debated (and will likely remain so, considering the lack of access to pre-malignant states in the brain). For this reason, the authors should use caution and simply refer to “malignant cells”. I agree that several papers, like the ref 131 listed by the authors (line 320), provide evidence supporting neural stem cells as the cell of origin for some GBM. Conversely, several other papers report that any type of neural cells, from the most undifferentiated to the most differentiated one can give rise to GBM like tumors (see for examples, Liu et al., Mosaic Analysis with Double Markers Reveals Tumor Cell of Origin in Glioma, Cell 2011; Galvao et al, Transformation of quiescent adult oligodendrocyteprecursor cells into malignant glioma through a multistep reactivation process, PNAS, 2014; Chow et al, Cooperativity within and among Pten, p53, and Rb Pathways Induces High-Grade Astrocytoma in Adult Brain, Cell, 2011; Friedman-Morvinski et al, Dedifferentiation of Neurons and Astrocytes by Oncogenes Can Induce Gliomas in Mice, Science, 2012; Weng et al, Single-Cell Transcriptomics Uncovers Glial Progenitor Diversity and Cell Fate Determinants during Development and Gliomagenesis, Cell, 2019).
- The brevity of the paragraphs devoted to the role of tumor-derived EVs in glioma is somewhat surprising with respect to the title of the review, although it might stem from the fact that the study of EV in glioma is recent. Nevertheless, paragraph 4.2 dealing with EV role in GBM could be enriched, at least for the part regarding the role of EV in interactions between tumor cells. The authors focused their discussion on examples of pro-tumorigenic effects of EVs. Recent studies have also provided evidences for EV anti-tumorigenic effects, notably through their cargos of non-coding RNAs. I am aware of two such examples. MiR320 was found to be released in EV by differentiated human GBM cells, to be expressed in situ by poorly aggressive GBM cells within patients’ tumors and to repress tumor cell growth in vivo using brain xenograft models (Fareh M, et al. Cell-based therapy using miR-302-367 expressing cells represses glioblastoma growth. Cell Death Dis. 2017). Long non-coding RNA SBF2-AS1 was reported to be upregulated in temozolomide -resistant GBM cells and tissues, and to promote resistance to temozolomide, a drug included in the standard treatment of GBM patients (Exosomal transfer of long non-coding RNA SBF2-AS1 enhances chemoresistance to temozolomide in glioblastoma. J Exp Clin Cancer Res. 2019).
- Precisions regarding the experimental models used to obtain the findings discusses in the review would be important to provide. Broad variations in pathological behavior and their associated molecular substrates can be expected between differing glioma models (e.g. human versus mouse GBM cells, serum versus serum-free GBM cell cultures, in vitro versus in vivo).
- line 341: Al-nedawi = Al-Nedawi
line 373 : significance of the abbreviation HOG ?
line 407 : may by physically » = may be physically
Author Response
Please note we have made minor revisions to the abstract for clarity (highlighted) and addressed the reviewer’s comments:
Reviewer #2:
(1) A schematic representation of the molecular pathways involved in EV biogenesis (described in paragraph 2.3) would be welcome to complete figure 1.
Response: The figure has been modified as suggested.
(2) The reference list contains numerous reviews. Although the authors justify their choice to refer to previous reviews in several instances throughout the article, they fail to do so in several other instances. In those cases, the references of the articles reporting the original findings have to be added, whenever possible.
Response: The references have been edited throughout to include ~27 original papers, wherever possible. In the scenario where original findings were added, referencing of reviews “as reviewed in-“ or “summarized in-“ has been stated.
(3) References 121 and 122 cited in regard of glioma gene expression profiles (lines 286-287) are 20 year-old. Tremendous progress having been achieved during the past years, more recent reviews have to be cited.
Response: The lines using these references have been modified as follows, with an updated reference list: “Gliomas are primary brain tumors comprised of tumor cells with gene expression profiles and morphological characteristics similar to glial cells (e.g. astrocytes, oligodendrocytes), as recently reviewed [119-121]. Glioma subtypes include oligodendroglioma, astrocytoma, glioblastoma, ependymoma, schwannoma and neurofibroma [119,122].” (line# 291-294; reference# 119-122)
- Perry, A.; Wesseling, P. Histologic classification of gliomas. Handb Clin Neurol 2016, 134, 71-95, doi:10.1016/B978-0-12-802997-8.00005-0.
- Crespo, I.; Vital, A.L.; Gonzalez-Tablas, M.; Patino Mdel, C.; Otero, A.; Lopes, M.C.; de Oliveira, C.; Domingues, P.; Orfao, A.; Tabernero, M.D. Molecular and Genomic Alterations in Glioblastoma Multiforme. Am J Pathol 2015, 185, 1820-1833, doi:10.1016/j.ajpath.2015.02.023.
- Wesseling, P.; van den Bent, M.; Perry, A. Oligodendroglioma: pathology, molecular mechanisms and markers. Acta Neuropathol 2015, 129, 809-827, doi:10.1007/s00401-015-1424-1.
- Louis, D.N.; Perry, A.; Reifenberger, G.; von Deimling, A.; Figarella-Branger, D.; Cavenee, W.K.; Ohgaki, H.; Wiestler, O.D.; Kleihues, P.; Ellison, D.W. The 2016 World Health Organization Classification of Tumors of the Central Nervous System: a summary. Acta Neuropathol 2016, 131, 803-820, doi:10.1007/s00401-016-1545-1.
(4) Line 126, the authors wrote: “some (pro-neural GBMs) feature IDH1/2 mutations and a hypermethylator phenotype”. In GBMs, IDH1/2 mutations are detected only in so-called secondary GBMs that evolve from lower grade lesions. Secondary GBMs represent a minority of GBMs. This point should be clarified.
Response: The text has been modified to: “However, not all pro-neural GBMs harbor PDGFRA alterations; some feature IDH1/2 mutations [129]. However, IDH mutations are only detected in secondary GBMs (arising from grade II and III gliomas) which constitute ~5% of the GBM cohort [131-133]. “(line #310-312).
(5) “The authors refer to tumor cells in glioma as “malignant glial cells” (lines 13, 147). The lineage identity of the cell(s) at the origin of glioma remains unknown, and is still debated (and will likely remain so, considering the lack of access to pre-malignant states in the brain). For this reason, the authors should use caution and simply refer to “malignant cells”. I agree that several papers, like the ref 131 listed by the authors (line 320), provide evidence supporting neural stem cells as the cell of origin for some GBM. Conversely, several other papers report that any type of neural cells, from the most undifferentiated to the most differentiated one can give rise to GBM like tumors (see for examples, Liu et al., Mosaic Analysis with Double Markers Reveals Tumor Cell of Origin in Glioma, Cell 2011; Galvao et al, Transformation of quiescent adult oligodendrocyteprecursor cells into malignant glioma through a multistep reactivation process, PNAS, 2014; Chow et al, Cooperativity within and among Pten, p53, and Rb Pathways Induces High-Grade Astrocytoma in Adult Brain, Cell, 2011; Friedman-Morvinski et al, Dedifferentiation of Neurons and Astrocytes by Oncogenes Can Induce Gliomas in Mice, Science, 2012; Weng et al, Single-Cell Transcriptomics Uncovers Glial Progenitor Diversity and Cell Fate Determinants during Development and Gliomagenesis, Cell, 2019).”
Response: We thank the reviewer for pointing out this misconception and for providing a list of important references to cite. In agreement with the reviewer’s suggestion, we have removed the word ‘glial’ when describing glioma cells and now call them “malignant cells” only, including in the summary and on line #179). To address the second part of this comment, we have described the other cellular sources implicated in gliomagenesis, as suggested (line #324-331). “GBMs are Grade IV astrocytomas and represent the most lethal brain tumor subtype [124]. NSCs residing in the SVZ carrying driver mutations (TERT promoter mutation, EGFR, PTEN and TP53 mutation) have been reported as the cell of origin for GBM [137]. However, other studies have found that any type of neural cell, including hippocampal NSCs [138], adult neurons and astrocytes [138,139], OPCs [140,141] and OPC intermediates (expressing low Pdgfra and high Olig1/2 levels) [142] can also give rise to GBM-like tumors. Interestingly, human GBM cell-derived EVs can drive transformation of human NSCs towards a tumorigenic state in vitro, highlighting the potential importance of GBM EVs in tumorigenesis [143]”
(6) “ The brevity of the paragraphs devoted to the role of tumor-derived EVs in glioma is somewhat surprising with respect to the title of the review, although it might stem from the fact that the study of EV in glioma is recent. Nevertheless, paragraph 4.2 dealing with EV role in GBM could be enriched, at least for the part regarding the role of EV in interactions between tumor cells. The authors focused their discussion on examples of pro-tumorigenic effects of EVs. Recent studies have also provided evidences for EV anti-tumorigenic effects, notably through their cargos of non-coding RNAs. I am aware of two such examples. MiR320 was found to be released in EV by differentiated human GBM cells, to be expressed in situ by poorly aggressive GBM cells within patients’ tumors and to repress tumor cell growth in vivo using brain xenograft models (Fareh M, et al. Cell-based therapy using miR-302-367 expressing cells represses glioblastoma growth. Cell Death Dis. 2017). Long non-coding RNA SBF2-AS1 was reported to be upregulated in temozolomide -resistant GBM cells and tissues, and to promote resistance to temozolomide, a drug included in the standard treatment of GBM patients (Exosomal transfer of long non-coding RNA SBF2-AS1 enhances chemoresistance to temozolomide in glioblastoma. J Exp Clin Cancer Res. 2019).”
Response: We thank the reviewer for this useful information. In agreement with the reviewer’s suggestion, section 4.2 has been further expanded to include the studies highlighted above and the following eight others (line #344-384; reference #152,155-161).
“Apart from EGFRvIII, EVs released by GBM cells (including GBM cell lines and GBM patient derived glioma/cancer stem cell line (GSC/CSC)) were demonstrated to carry the chloride intracellular channel-1 (CLIC1) protein [156]. CLIC1 plays a role in cell cycle regulation [157] and has previously been implicated in GBM growth, such that high CLIC1 levels correlate to poor prognosis in GBM patients [158]. It was later demonstrated that treatment of GBM cells with EVs carrying CLIC1 (1µg/ml) resulted in increased GBM cell proliferation in vitro and in a mouse GBM xenograft model system in vivo [156].
High expression of miR-21 is a common feature in GBM patient tissues and established GBM cell lines in vitro, and suppression of mi-R21 in vitro results in decreased proliferation and increased apoptosis [159,160]. Additionally, GBM patients with high miR-21 levels exhibited poor prognosis [160]. Interestingly, miR-21 carried in sEVs in the CSF of GBM patients has also been reported, where the level of miR-21 in sEVs is directly related to the glioma status of the patient [154]. Thus, recurring GBM patients demonstrate low levels of sEV miR-21 post-surgical resection, compared to higher levels of miR-21 observed prior to surgical resection [154]. Hence, sEVs carrying miR-21 in the CSF are an excellent biomarker for checking the glioma status of GBM patients. Also, antisense miRNA oligonucleotides against miR-21 loaded onto sEVs have been recently tested as a delivery system in vivo using a mouse xenograft model for assessing their therapeutic potential [161]. On similar lines, miR-21 inhibition in GBM cells in vitro was introduced using sEVs engineered to carry an miR-21 sponge construct (i.e. three miR-21 complementary sequences joined by linker sequences) [162]. Using this approach suppression of miR-21 target genes, PDCD4 and RECK, in GBM cells was reverted along with an increase in apoptosis and decrease in cell proliferation, similar to prior studies [159,162]. Additionally, the introduction of sEV loaded miR-21 sponge constructs in a rat xenograft model of GBM led to a significant reduction in tumor volume compared to the control group [162]. Thus, miR-21 inhibition via sEVs is an active area of research for developing new therapeutic strategies.
Tumor repressive functions have also recently been attributed to GBM derived EVs. miR-302-367 cluster was found to repress stemness in GBM patient derived GSC lines [163]. Fareh et al. engineered GBM patient derived GSC lines to express miR-302-367 cluster, which then released sEVs carrying miR-302-367 as cargo. Transfer of miR-302-367 to recipient GSC lines via sEVs repressed the stem cell-like nature of GSCs, as demonstrated via an inhibition of cell stemness and proliferation marker expression (e.g. Shh, SOX2, Cyclin D, Cyclin A) [163]. Thus, miR-302-367 can block GBM growth in a paracrine fashion and miR-302-367 delivery via sEVs is currently being assessed as a potent therapeutic approach for GBM patients. Interestingly, human GBM cell-derived sEVs were reported to carry O-methylguanine-DNA methyltransferase (or MGMT) mRNA, which is an indicator of GBM drug resistance status [164]. Additionally, temozolomide (TMZ) resistant human GBM cells release sEVs carrying the lncRNA SBF2-AS1, which represses miR-151a-3p and XRC44 expression in vitro [29]. Transfer of lncRNA SBF2-AS1 via sEVs to neighboring GBM cells also endowed TMZ resistance in the recipient GBM cells [29]. Thus, lncRNA SBF2-AS1 in sEVs can be an excellent readout/biomarker of TMZ resistance in GBM patients. Thus, personalization of chemotherapeutic treatment may be possible using EV cargo as more precise readouts of the current glioma status [164].”
- Shi, R.; Wang, P.Y.; Li, X.Y.; Chen, J.X.; Li, Y.; Zhang, X.Z.; Zhang, C.G.; Jiang, T.; Li, W.B.; Ding, W., et al. Exosomal levels of miRNA-21 from cerebrospinal fluids associated with poor prognosis and tumor recurrence of glioma patients. Oncotarget 2015, 6, 26971-26981, doi:10.18632/oncotarget.4699.
- Setti, M.; Osti, D.; Richichi, C.; Ortensi, B.; Del Bene, M.; Fornasari, L.; Beznoussenko, G.; Mironov, A.; Rappa, G.; Cuomo, A., et al. Extracellular vesicle-mediated transfer of CLIC1 protein is a novel mechanism for the regulation of glioblastoma growth. Oncotarget 2015, 6, 31413-31427, doi:10.18632/oncotarget.5105.
- Valenzuela, S.M.; Mazzanti, M.; Tonini, R.; Qiu, M.R.; Warton, K.; Musgrove, E.A.; Campbell, T.J.; Breit, S.N. The nuclear chloride ion channel NCC27 is involved in regulation of the cell cycle. J Physiol 2000, 529 Pt 3, 541-552, doi:10.1111/j.1469-7793.2000.00541.x.
- Setti, M.; Savalli, N.; Osti, D.; Richichi, C.; Angelini, M.; Brescia, P.; Fornasari, L.; Carro, M.S.; Mazzanti, M.; Pelicci, G. Functional role of CLIC1 ion channel in glioblastoma-derived stem/progenitor cells. J Natl Cancer Inst 2013, 105, 1644-1655, doi:10.1093/jnci/djt278.
- Chan, J.A.; Krichevsky, A.M.; Kosik, K.S. MicroRNA-21 is an antiapoptotic factor in human glioblastoma cells. Cancer Res 2005, 65, 6029-6033, doi:10.1158/0008-5472.CAN-05-0137.
- Yang, C.H.; Yue, J.; Pfeffer, S.R.; Fan, M.; Paulus, E.; Hosni-Ahmed, A.; Sims, M.; Qayyum, S.; Davidoff, A.M.; Handorf, C.R., et al. MicroRNA-21 promotes glioblastoma tumorigenesis by down-regulating insulin-like growth factor-binding protein-3 (IGFBP3). J Biol Chem 2014, 289, 25079-25087, doi:10.1074/jbc.M114.593863.
- Kim, G.; Kim, M.; Lee, Y.; Byun, J.W.; Hwang, D.W.; Lee, M. Systemic delivery of microRNA-21 antisense oligonucleotides to the brain using T7-peptide decorated exosomes. J Control Release 2020, 317, 273-281, doi:10.1016/j.jconrel.2019.11.009.
- Monfared, H.; Jahangard, Y.; Nikkhah, M.; Mirnajafi-Zadeh, J.; Mowla, S.J. Potential Therapeutic Effects of Exosomes Packed With a miR-21-Sponge Construct in a Rat Model of Glioblastoma. Front Oncol 2019, 9, 782, doi:10.3389/fonc.2019.00782.
(7) Precisions regarding the experimental models used to obtain the findings discusses in the review would be important to provide. Broad variations in pathological behavior and their associated molecular substrates can be expected between differing glioma models (e.g. human versus mouse GBM cells, serum versus serum-free GBM cell cultures, in vitro versus in vivo).
Response: In agreement with the reviewer’s suggestion, specific details regarding experimental models involved has been incorporated throughout the paper wherever possible.
(8) line 341: Al-nedawi = Al-Nedawi
Response: The text has been corrected to represent the name correctly (line #388).
(9) line 373 : significance of the abbreviation HOG ?
Response: The text has been corrected to expand the abbreviation of HOG: Human oligodendroglioma cell line (line #418).
(10) line 407 : may by physically » = may be physically
Response: The text has been modified to correct the error (line #452). The sentence now reads “Liquid biopsies” based on the analysis of whole blood are based on the premise or assumption that the BBB within a brain tumor may be physically compromised and hence permissive to release or intravasation of biomarker material (proteins, cells, metabolites, DNA/RNA, extracellular vesicles, etc.) into the hematogenous circulation.”
Reviewer 3 Report
This review is very well written and exciting, covering all the latest findings in a rapidly moving field alongside demonstrating an excellent depth of knowledge. Right from the start it sets the reader up for an excellent read describing clearly the utility of exosomes as biomarkers for monitoring disease response and progression.
I have a few very minor suggested changes:
Figure 2 labels are too small
Line 445 lumbar punctures are in fact routinely performed in childhood brain cancer patients – please amend
Line 476- at an appropriate/matched? female: male ratio
Line 491- a few studies have been done on CSF from normal controls (sampling to detect something other than brain tumours). This approach should be considered and referenced
Line 555- in a similar manner?
Lines 556-569 discussed the application of mass cytometry but there are no associated references
Author Response
Please note we have made minor revisions to the abstract for clarity (highlighted) and addressed the reviewer’s comments:
Reviewer #3:
(1) Figure 2 labels are too small
Response: The figure has been modified as suggested (see response (1) to reviewer 1).
(2) Line 445 lumbar punctures are in fact routinely performed in childhood brain cancer patients – please amend
Response: The text has been modified and the new sentence now reads “However, it suffers from one major drawback: it can only be obtained via lumbar puncture, a technically challenging means of biofluid collection and is seldom performed in brain cancer patients for diagnostic purposes.” (line #488-490)
(3) Line 476- at an appropriate/matched? female: male ratio
Response: The text has been modified and the sentence now reads: “This would ideally comprise of patients that are age-matched to the disease of interest, are MRI scan negative, and at an appropriately matched female:male ratio.” (line #519)
(4) Line 491- a few studies have been done on CSF from normal controls (sampling to detect something other than brain tumours). This approach should be considered and referenced
Response: The text has been modified to incorporate two new references to support the statement (line #535; reference#203,204). The sentence now reads ”While these “housekeeping” biomarkers already exist for EV analysis in plasma/serum/urine, it is not clear what biomarkers should be used for CSF [203,204].”
The new references are:
- Akers, J.C.; Ramakrishnan, V.; Yang, I.; Hua, W.; Mao, Y.; Carter, B.S.; Chen, C.C. Optimizing preservation of extracellular vesicular miRNAs derived from clinical cerebrospinal fluid. Cancer Biomark 2016, 17, 125-132, doi:10.3233/CBM-160609.
- Akers, J.C.; Hua, W.; Li, H.; Ramakrishnan, V.; Yang, Z.; Quan, K.; Zhu, W.; Li, J.; Figueroa, J.; Hirshman, B.R., et al. A cerebrospinal fluid microRNA signature as biomarker for glioblastoma. Oncotarget 2017, 8, 68769-68779, doi:10.18632/oncotarget.18332.
(5) Line 555- in a similar manner?
Response: The text has been modified (line #599), and the sentence now reads as ” Fluorescent negative stains such as high-molecular weight dextrans conjugated to fluorophores could be used to image EVs via super resolution microscopy in a similar manner to how osmium tetroxide is used as a negative stain for imaging EVs via transmission electron microscopy.”
(6) Lines 556-569 discussed the application of mass cytometry but there are no associated references
Response: The text has been modified to incorporate three new references to support the statement (line #608,612; reference #208-210). The sentences now read as ” The lack of background or “bleed through” with rare earth metal ions as the “fluorophore” in mass cytometry also offers a significant advantage in that if a rare earth metal signal is detected on an event, there is absolute certainty that it is a true signal and not an artefact [208,209]. Fluorophores suffer from noise and spectral overlaps between the dyes commonly used in flow cytometry, thus introducing doubt in any signal observed on an event, whether it be a cell or EV. The opportunity to use mass cytometry for EV analysis means that a very high number of EV sub-populations can be enumerated in a given sample, allowing for quantitation of highly specific subsets of EVs or quantitation of all possible blood cell EV subtypes in any plasma sample [210].”
The new references are:
- Bendall, S.C.; Simonds, E.F.; Qiu, P.; Amir el, A.D.; Krutzik, P.O.; Finck, R.; Bruggner, R.V.; Melamed, R.; Trejo, A.; Ornatsky, O.I., et al. Single-cell mass cytometry of differential immune and drug responses across a human hematopoietic continuum. Science 2011, 332, 687-696, doi:10.1126/science.1198704.
- Bandura, D.R.; Baranov, V.I.; Ornatsky, O.I.; Antonov, A.; Kinach, R.; Lou, X.; Pavlov, S.; Vorobiev, S.; Dick, J.E.; Tanner, S.D. Mass cytometry: technique for real time single cell multitarget immunoassay based on inductively coupled plasma time-of-flight mass spectrometry. Anal Chem 2009, 81, 6813-6822, doi:10.1021/ac901049w.
- Ornatsky, O.; Bandura, D.; Baranov, V.; Nitz, M.; Winnik, M.A.; Tanner, S. Highly multiparametric analysis by mass cytometry. J Immunol Methods 2010, 361, 1-20, doi:10.1016/j.jim.2010.07.002.